# Birth month as a risk factor of allergic diseases: Analysis of database of about 50 thousand children

Alexander Pampura[1,2], Natalia Esakova[2]*, Daria Dolotova[2], Olga Serebryakova[3‡], Nikita Chikunov[4‡]

1 Morozov Children's City Hospital, Moscow, Russian Federation, 2 Veltischev Research and Clinical Institute for Pediatrics and Pediatric Surgery of the Pirogov Russian National Research Medical University of the Russian Ministry of Health, Moscow, Russian Federation, 3 Pirogov Russian National Research Medical University of the Russian Ministry of Health, Moscow, Russian Federation, 4 Moscow Center for Innovative Technologies in Healthcare, Moscow, Russian Federation

☙ These authors contributed equally to this work.
‡ OS and NC also contributed equally to this work.
* env007@rambler.ru

## Abstract

### Background

The season of birth is a factor influencing the child during the neonatal adaptation period and potentially affecting the risk of allergies. The objective of this study was to ascertain the association between the month of birth and the subsequent development of allergic diseases in Moscow children, Russia.

### Methods

in 2024 the de-identified data from medical records and parental questionnaires of 49,857 children under the age of 18 was retrieved from the Moscow Unified Medical Information and Analytical System. The database contained the information regarding the presence of atopic dermatitis, asthma, allergic rhinitis, age, sex and family history of allergies. The statistical processing involved the calculation of crude odds ratio (cOR), adjusted odds ratio (aOR) based on multivariate logistic regression.

### Results

the odds of allergic rhinitis among children born between October and April was found to be significantly higher in comparison to July (reference), with the strongest association observed for December (aOR, 1,342; 95% CI, 1,203−1,497), January (aOR, 1,386; 95% CI, 1,243−1,546) and February (aOR, 1,371; 95% CI, 1,226−1,533). In these months, the odds were 34−38% higher than in July. The odds of atopic dermatitis among children born between August and February was

**Data availability statement:** The data cannot be openly accessible due to legal restrictions. The Moscow Department of Health legally owns the data of this study. Access to the database can be requested through the official email address from the Local Ethics Committee at Veltischev Research and Clinical Institute for Pediatrics and Pediatric Surgery: doc@pedklin.ru.

**Funding:** The study was sponsored by Moscow Center for Innovative Technologies in Healthcare and was funded by grant from the Moscow government [research project No. 2412-41/22].

**Competing interests:** The authors have declared that no competing interests exist.

significantly higher compared to April (reference), the greatest association observed for October (aOR, 1,169; 95% CI, 1,059−1,291), with the association being 16% higher than for April.

## Conclusion

This is the first study in Russia to demonstrate that children born in October in Moscow face elevated odds of atopic dermatitis, while children born in December, January, and February are more susceptible to allergic rhinitis. The association detected was independent of sex, age, family allergic history and combination of allergic diseases, which merits further investigation.

## Introduction

Atopic dermatitis (AD), allergic rhinitis (AR), and asthma are the most prevalent allergic diseases in children, frequently demonstrating an age-dependent sequence of sensitization and clinical manifestations, termed "the Atopic March". AD commonly manifests in early childhood, preceding the development of AR and asthma. This observation indicates the pathogenetic relationship between these diseases, highlighting the importance of comprehensive investigations of potential risk factors for their development in the pediatric population [1–3].

A substantial body of research has revealed associations between male sex, family allergy history, characteristics of the causative allergen, genetic, environmental, infectious and other factors and the risk of AD, AR and asthma development [4–9]. A growing amount of research considers the role of the birth season as an important factor with a significant impact on the child during the neonatal adaptation period and potentially influencing the risk of developing allergic diseases. Thus, several studies have identified a heightened risk of developing AD, AR and asthma. However, the findings are not universally consistent, with some studies reporting no clear associations [10–19]. Notably, the season factor itself is heterogeneous, encompassing numerous uncontrolled and controlled factors. Such factors include air temperature and humidity, solar activity, the specificity and degree of allergen exposure (e.g., pollen, dust) in different seasons, the infectious burden, the seasonal variations in the nutrition of the nursing mothers, etc. For example, the winter months are characterized by minimal solar activity and, as a result, decreased vitamin D levels, as well as high exposure to house dust mites and epidermal allergens in homes, which can potentially increase odds of allergy among children born during this period. The possible roles of increased cord blood IgE levels in winter, increased immune activity in general in winter months, including gene expression of several interleukins are discussed [19]. Consequently, the impact of the season of birth on the risk of developing allergic diseases may exhibit notable variations across different regions, even within the same country, owing to the unique characteristics of each region.

A comprehensive understanding of the mechanisms and extent of the influence of the season of birth on the risk of allergic diseases is imperative for the

optimization of preventive measures during pregnancy planning, as well as prior to and following childbirth. To date, few studies worldwide have examined this relationship, with no such studies having been conducted in Russia. The objective of this study was to ascertain the potential association between the month of birth and the odds of developing the most prevalent allergic diseases: AD, AR, and asthma, among children living in Moscow (central region of the Russian Federation).

## Methods

The study was conducted in 2024 and included an analysis of the data from 49,857 children aged 0–18 years, retrieved from the Moscow Unified Medical Information and Analytical System (EMIAS). EMIAS is a modern information platform in Moscow, which contains electronic medical records available to the patient and the doctor. At each visit of a patient, the doctor (outpatient or inpatient) enters information into the EMIAS medical record, indicating the reason for the visit; in case of a disease, the diagnosis is specified; the same system displays appointments and results of laboratory and instrumental tests. All necessary information, including the diagnosis code according to ICD (International Classification of Diseases, 10th revision), is entered into the EMIAS by healthcare professionals. All patients (patient parents) provide digital informed consent upon first registration in the EMIAS. It is important to note that in Russia, the basic medical care for children and adults is provided through compulsory health insurance, i.e., free of charge. Children may visit a pediatrician due to illness (e.g., often in the case of an acute viral infection or in connection with some specific complaints or diseases), or patients may come without complaints or diseases, e.g., to obtain school or sectional certificates, to undergo a checkup or vaccination. Thus, the study population included both children with diseases and healthy children who visited a pediatrician. The exported dataset contained data exclusively from children whose parents had voluntarily completed a survey questionnaire (integrated within EMIAS) between February 2024 and November 2024 at a pediatrician's office. The parental questionnaire aimed to ascertain the existence of a family allergy history. Thus, the main inclusion criteria for the formation of the database from EMIAS were: age under 18 years and a parentally fully completed questionnaire to identify family allergy history. We excluded all cases in which at least one of the study variables was missing, so there were no missing values in the dataset. All offload data underwent depersonalization and included medical records (both outpatient and inpatient) as well as parental questionnaire results.

The resulting dataset was searched for patients with established allergic diseases consistent with ICD-10 diagnosis codes as follows: L20.0, L20.8, L20.9 – AD; J30.1, J30.2, J30.3 –AR; and J45.0, J45.1, J45.8, J45.9 - asthma. To account for the influence of possible confounders on the association of these diseases with the month of birth, information on the age and sex of the child, as well as on the presence of family allergic history, was included in the dataset. The study was approved by the Local Ethics Committee at Veltischev Research and Clinical Institute for Pediatrics and Pediatric Surgery (Protocol №1, dated January 19, 2024). All patients (patient parents) provide digital informed consent.

The data were assessed for normality of the distribution (Shapiro–Wilk test). The distribution of the patients' age was described via medians and quartiles (Me [Q1; Q3]), given that the distribution of the data deviated from normal. Qualitative variables were described using absolute and relative frequencies. To examine the effects of birth month and season on the odds of developing allergic disease, crude and adjusted odds ratios (cOR and aOR, respectively) were calculated, for which both point estimates and 95% confidence interval (95% CI) were indicated. The influence of confounders on the aOR calculation was evaluated via multivariate logistic regression [20]. The Benjamini-Hochberg procedure was employed to address multiple comparisons, leading to an adjusted significance threshold level of 0.0281 [21]. The results are presented via bar and forest plots, and the analysis was conducted in the RStudio environment, leveraging the tidymodels and ggplot2 packages. In all the comparisons, either the period or the month of birth with the lowest incidence of the disease was used as a reference.

## Results

### Characteristics of the pediatric patients in Moscow

The final analysis included data from medical records of 49,857 patients aged 0–18 years (median age 7 [4;10] years), among whom 43.8% were female and 56.2% were male. The frequency of birth by month varied from 7.6% to 8.9%. The prevalence of AD was 29.4%, AR – 23.2%, asthma – 4.8%, family history of allergic disease was established in 50.9% of the children (Table 1).

### Association between the birth month/season and the odds of developing an allergic disease

For initial crude estimation of the association between the birth season and the odds of developing an allergic disease we categorized children's birth seasons into those with high vs low prevalence of allergic disease compared with the yearly mean values. This resulted in periods of birth for children with high and low risk of disease. A high prevalence of AD was observed among patients born from September through February, of AR – from November through March, of asthma – from August through March (Fig 1).

The proportions of patients born during high- and low- risk periods were further compared, and the comparisons revealed statistically significant differences (Table 2). In each comparison, the period with the lowest prevalence of the disease was used as a reference.

**Table 1. Characteristics of the pediatric patients in Moscow.**

| Characteristics of children n = 49 857 | n (%) or Me [Q1; Q3] |
|---|---|
| **Age (years)** | 7 [4;10] |
| **Sex** | |
| Male | n = 28,047 (56.2%) |
| Female | n = 21,810 (43.8%) |
| **Month of birth** | |
| January | n = 4,206 (8.4%) |
| February | n = 3,779 (7.6%) |
| March | n = 4,128 (8.3%) |
| April | n = 3,823 (7.7%) |
| May | n = 3,948 (7.9%) |
| June | n = 4,209 (8.4%) |
| July | n = 4,452 (8.9%) |
| August | n = 4,416 (8.9%) |
| September | n = 4,243 (8,5%) |
| October | n = 4,325 (8.7%) |
| November | n = 4,104 (8.2%) |
| December | n = 4,224 (8.5%) |
| **Allergic diseases** | |
| Atopic dermatitis | n = 14,650 (29.4%) |
| Allergic rhinitis | n = 11,566 (23.2%) |
| Asthma | n = 2,415 (4.8%%) |
| **Family history of allergy** | |
| Present | n = 25,401 (50.9%) |
| Absent | n = 24,456 (49.1%) |

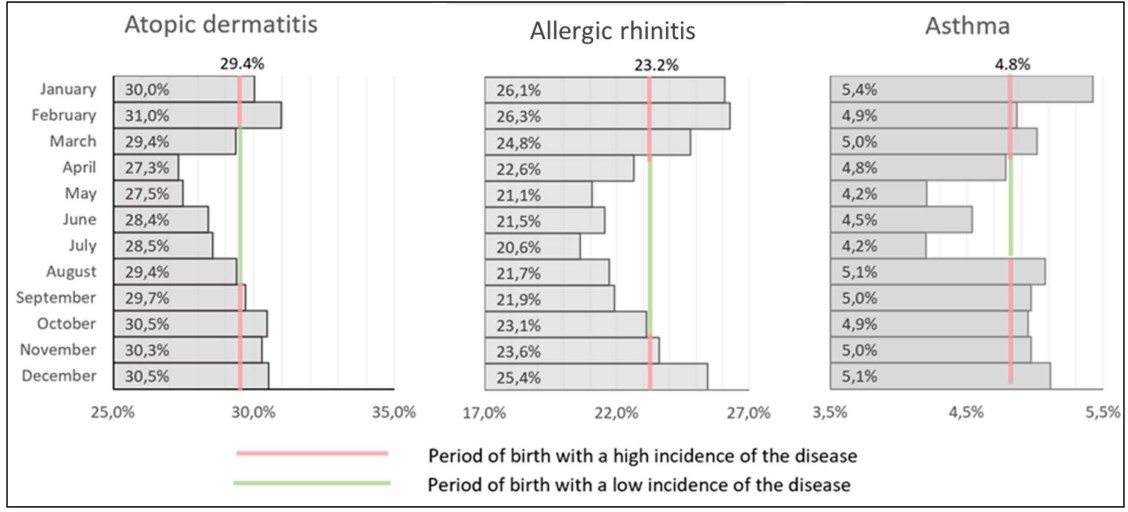

**Fig 1. Prevalence of allergic disease depending on the month of birth in children: categorization of high vs low seasons compared to yearly mean values.**

*Reference – period with the lowest prevalence of the disease. Abbreviations: cOR- crude odds ratio; aOR- adjusted odds ratio; 95% CI – 95% confidence interval. Adjusted for periods of birth associated with high/low risk of disease, sex, age, family allergy history, the presence of allergic disease of interest -AD, AR, asthma – in combinations. Statistically significant differences are highlighted in bold. \*p < 0.05; \*\*p < 0.0281 (with Benjamini Hochberg adjustment).*

The analysis of OR revealed the strongest association between the month/season of birth and the odds of developing AR. Furthermore, the odds of developing asthma among patients born between August and March was 10% higher (aOR, 1.100; 95% CI, 1.003–1.207) compared to those born in April through July (reference). In the case of AD, the association between birth month/season and the development of AD was less significant. Specifically, children born between September and February demonstrated a 1.075-fold increase in the likelihood of developing AD (aOR, 95% CI, 1.033–1.119) compared to those born between March and August (reference). The robustness of the observed seasonal effect on the odds of developing AD and AR was maintained when accounting for multiple comparisons (Table 2, Fig 2).

In the study population, male pediatric patients represented the majority of cases, with proportions ranging from 59% to 72.5% among the specific diseases. The analysis revealed that children with asthma presented the highest prevalence of a family allergy history, reaching 72.5%. This prevalence was followed by the group of patients with AR and AD, where it amounted to 68.5% and 61.6%, respectively. The OR evaluation demonstrated that male patients and children with a family allergy history presented a significantly greater odds of developing the allergic disease under consideration than female patients and children without a family allergy history, respectively. The strongest associations between disease development and demographic factors were observed for patients with asthma (male sex: aOR, 1.740; 95% CI, 1.584–1.912; family allergy history: aOR, 1.829; 95% CI, 1.663–2.012) and AR (male sex: aOR, 1.636; 95% CI, 1.561–1.714; family allergy history: aOR, 2.186; 95% CI, 2.085–2.291). The weakest associations between the disease development and demographic factors were observed for patients with AD (male sex: aOR, 1.042; 95% CI, 1.000–1.085; family allergy history: aOR, 1.568; 95% CI, 1.505–1.633). In general, we observed that the association between the odds of developing AR, asthma and AD and the demographic parameters (family allergy history, sex of the patient) were stronger than those observed for the birth month/season. Despite this observation, the association between the birth month/season and the odds of developing AD, AR and asthma remained statistically significant following the OR analysis with consideration of confounders (Table 2, Fig 2).

**Table 2. Odds ratios for the risk of allergic disease in children during the period of birth associated with high risk, sex, age, family allergy history, presence of allergic disease AD, AR, asthma.**

| | Parameter | Allergic disease is present, n (%) or Me [Q1; Q3] | Allergic disease is absent, n (%) or Me [Q1; Q3] | cOR (95% CI) | aOR (95% CI) |
|---|---|---|---|---|---|
| **Atopic dermatitis (n = 14650)** | Period of birth associated with high risk (September through February) | 30.3% (n = 7545) | 28.4% (n = 7105) | **1.095 (1.053-1.138)** p<<0.001** | **1.075 (1.033-1.119)** p<<0.001** |
| | Male sex | 59% (n = 8646) | 55.1% (n = 19401) | **1.173 (1.128-1.220)** p<<0.001** | **1.042 (1.000-1.085)** p = 0.049* |
| | Family allergy history | 61.6% (n = 9027) | 46.5% (n = 16387) | **1.844 (1.773−1.918)** p<<0.001** | **1.568 (1.505-1.633)** p<<0.001** |
| | AR | 37.1% (n = 5442) | 17.4% (n = 6124) | **2.807 (2.688-2.931)** p<<0.001** | **2.335 (2.229 −2.446)** p<<0.001** |
| | Asthma | 7.8% (n = 1138) | 3.6% (n = 1277) | **2.236 (2.061-2.430)** p<<0.001** | **1.384 (1.269 −1.510)** p<<0.001** |
| | Age, years | 7 [5; 11] | 7 [3; 10] | **1.054 (1.049-1.059)** p<<0.001** | **1.023 (1,018-1.028)** p<<0.001** |
| **Allergic rhinitis (n = 11566)** | Period of birth associated with high risk (November through March) | 25.2% (n = 5156) | 21.8% (n = 6410) | **1.211(1.161-1.263)** p<<0.001** | **1.225 (1.170-1.282)** p<<0.001** |
| | Male sex | 66.4% (n = 7684) | 53.2% (n = 20363) | **1.743 (1.669-1.820)** p<<0.001** | **1.636 (1.561-1.714)** p<<0.001** |
| | Family allergy history | 68.5% (n = 7922) | 4.7% (n = 17492) | **2.585 (2.474-2.701)** p<<0.001** | **2.186 (2.085-2.291)** p<<0.001** |
| | AD | 47.1% (n = 5442) | 24% (n = 9208) | **2.807 (2.688-2.931)** p<<0.001** | **2.410 (2.301-2.525)** p<<0.001** |
| | Asthma | 12.9% (n = 1478) | 2.4% (n = 928) | **5.940 (5.457-6.466)** p<<0.001** | **3.428 (3.130-3.753)** p<<0.001** |
| | Age, years | 9 [6; 12] | 6 [3; 10] | **1.168 (1.162-1.175)** p<<0.001** | **1.156 (1.149-1.163)** p<<0.001** |
| **Asthma (n = 2415)** | Period of birth associated with high risk (August through March) | 5,1% (n = 1688) | 4.4% (n = 727) | **1.149 (1.051-1.256)** p<<0.001** | **1.100 (1.003-1.207)** p = 0.043* |
| | Male sex | 72.5% (n = 1751) | 55.4% (n = 26296) | **2.121 (1.936-2.323)** p<<0.001** | **1.740 (1.584-1.912)** p<<0.001** |
| | Family allergy history | 72,54% (n = 1752) | 49.9% (n = 23662) | **2.656 (2.424−2.909)** **p<<0.001** | **1.829 (1.663–2.012)** p<<0.001** |
| | AD | 47.1% (n = 1138) | 28.5% (n = 13512) | **2.238 (2.061-2.430)** p<<0.001** | **1.491 (1.367-1.627)** p<<0.001** |
| | AR | 61.6% (n = 1478) | 21.2% (n = 10079) | **5.940 (5.457-6.466)** p<<0.001** | **3.530 (3.255-3.863)** p<<0.001** |
| | Age, years | 10 [7; 12] | 7 [4; 10] | **1.198 (1.184-1.211)** p<<0.001** | **1.161 (1.147-1.176)** p<<0.001** |

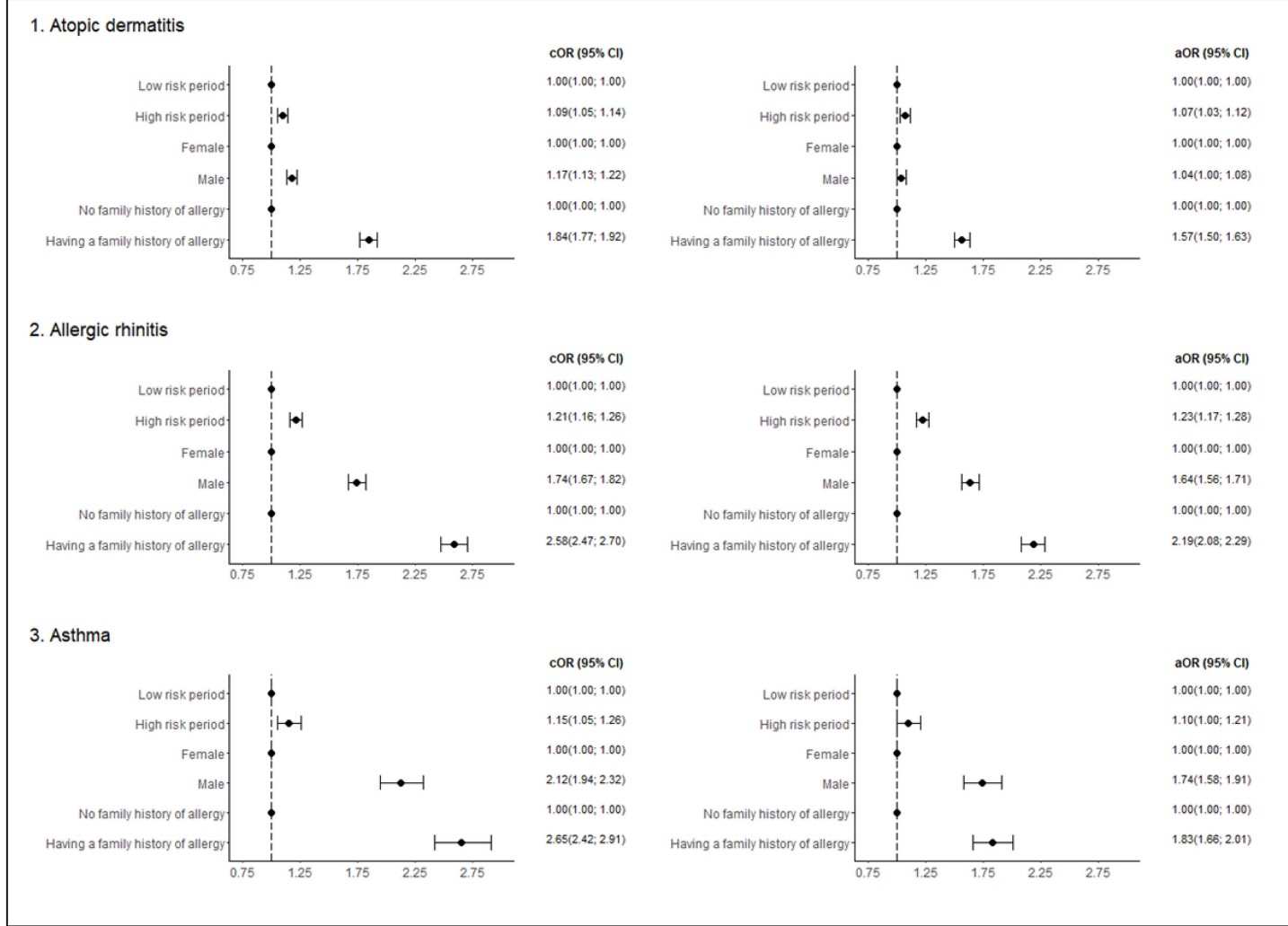

**Fig 2. Periods of birth of children with high and low risk of allergic disease. Odds ratio and 95% CI for outcomes (development of allergic disease) depending on the period of birth, sex, and family history of allergy.** *Reference – period with the lowest prevalence of the disease. Abbreviations: cOR- crude odds ratio; aOR- adjusted odds ratio; 95% CI – 95% confidence interval. Adjusted for periods of birth associated with high/low risk of disease, sex, age, family allergy history, the presence of allergic disease of interest -AD, AR, asthma – in combinations.*

To perform a more comprehensive evaluation, we conducted a study to ascertain the association between the specific birth month of the children and the subsequent development of allergic disease (Table 3, Fig 3). In all comparisons, the month of birth with the lowest disease prevalence determined in the initial crude estimate was used as the reference (Fig 1).

Analysis of the OR revealed an association between the specific month of birth and the odds developing AR and AD. In all the comparisons, the month of birth with the least incidence of the disease was used as a reference. In particular, the individuals born between October and April presented a substantially greater odds of developing AR compared with those born in July (reference), with the strongest associations observed in December (aOR, 1.342; 95% CI, 1.203–1.497), January (aOR, 1.386; 95% CI, 1.243–1.546), and February (aOR, 1.371; 95% CI, 1.226–1.533). The odds of AR development in these months were 34–38% greater than that in July, and the significance of the association was confirmed via OR analysis when the adjustment for multiple comparisons was taken into account (Table 3, Fig 3).

**Table 3. Odds ratio for the risk of developing allergic disease in children born in different months of the year.**

| | Birth Month | Allergic disease is present, n (%) | Allergic disease is absent, n (%) | cOR (95% CI) | aOR (95% CI) |
|---|---|---|---|---|---|
| **Atopic dermatitis** | January | 8.6% (n=1263) | 8.4% (n=2943) | **1,**142 (1,037-1,**259)** p=0.007** | **1,**113 (1,007-1,**231)** p=0.035* |
| | February | 8% (n=1171) | 7.4% (n=2608) | **1,195** (1,082−1,**320)** p<<0.001** | **1,155** (1,042-1,**279)** p=0.006** |
| | March | 8.3% (n=1212) | 8.3% (n=2916) | **1,**106 (1,003-1,**220)** p=0.043* | 1,088 (0,984−1,203) p=0.101 |
| | April | 7.1% (n=1044) | 7.9% (n=2779) | Reference | Reference |
| | May | 7.4% (n=1085) | 8.1% (n=2863) | 1,009 (0,913−1,115) p=0.860 | 1,027 (0,927−1,138) p=0.610 |
| | June | 8.2% (n=1195) | 8.6% (n=3014) | 1,055 (0,957−1,164) p=0.280 | 1,067 (0,965−1,180) p=0.207 |
| | July | 8.7% (n=1271) | 9.0% (n=3181) | 1,064 (0,966−1,171) p=0.210 | 1,094 (0,990−1,208) p=0.077 |
| | August | 8.9% (n=1298) | 8.9% (n=3118) | **1,10**8 (1,006-1,**220)** p=0.036* | **1,117** (1,012-1,**234)** p=0.028** |
| | September | 8.6% (n=1260) | 8.5% (n=2983) | **1,124** (1,020-1,**239)** p=0.018** | **1,140** (1,031-1,**259)** p=0.01** |
| | October | 9% (n=1318) | 8.5% (n=3007) | **1,167** (1,060−1,**285)** p=0.002** | **1,169** (1,059−1,**291)** p=0.002** |
| | November | 8.5% (n=1243) | 8.1% (n=2861) | **1,156** (1,049−1,**275)** p=0.003** | **1,155** (1,045-1,**277)** p=0.005** |
| | December | 8.8% (n=1290) | 8.3% (n=2934) | **1,170** (1,062−1,**289)** p=0.001** | **1,152** (1,043-1,**273)** p=0.005** |
| **Allergic rhinitis** | January | 9.5% (n=1097) | 8.1% (n=3109) | **1,358** (1,2**29−1,501)** p<<0.001** | **1,386** (1,243−1,**546)** p<<0.001** |
| | February | 8.6% (n=993) | 7.3% (n=2786) | **1,372** (1,238−**1,520)** p<<0.001** | **1,371** (1,226−**1,533)** p<<0.001** |
| | March | 8.8% (n=1023) | 8.1% (n=3105) | **1,268** (1,**146−1,404)** p<<0.001** | **1,276** (1,**143−1,424)** p<<0.001** |
| | April | 7.5% (n=865) | 7.7% (n=2958) | **1,**126 (1,014-1,**250)** p=0.027** | **1,14**9 (1,025-1,**288)** p=0.017** |
| | May | 7.2% (n=832) | 8.1% (n=3116) | 1,028 (0,925−1,142) p=0.609 | 1,043 (0,931−1,170) p=0.466 |
| | June | 7.8% (n=907) | 8.6% (n=3302) | 1,057 (0,954 −1,173) p=0.289 | 1,055 (0,943−1,180) p=0.347 |
| | July | 7.9% (n=918) | 9.2% (n=3534) | Reference | Reference |
| | August | 8.3% (n=959) | 9.0% (n=3457) | 1,068 (0,964−1,183) p=0.206 | 1,039 (0,931−1,161) p=0.494 |
| | September | 8% (n=929) | 8.7% (n=3314) | 1,079 (0,974−1,196) p=0.146 | 1,070 (0,957−1,196) p=0.237 |
| | October | 8.6% (n=1000) | 8.7% (n=3325) | **1,158** (1,046-1,**281)** p=0.005** | **1,**147 (1,027-1,**280)** p=0.015** |
| | November | 8.4% (n=969) | 8.2% (n=3135) | **1,190** (1,074−1,**318)** p=0.001** | **1,183** (1,059−1,**323)** p=0.003** |
| | December | 9.3% (n=1074) | 8.2% (n=3150) | **1,313** (1,187−1,**451)** p<<0.001** | **1,342** (1,203−1,**497)** p<<0.001** |

*(Continued)*

**Table 3.** (Continued)

| | Birth Month | Allergic disease is present, n (%) | Allergic disease is absent, n (%) | cOR (95% CI) | aOR (95% CI) |
|---|---|---|---|---|---|
| Asthma | January | 9.4% (n = 228) | 8.4% (n = 3978) | **1,307 (1,072−1,594)** p = 0.008** | 1,218 (0,992−1,496) p = 0.06 |
| | February | 7.6% (n = 184) | 7.6% (n = 3595) | 1,167 (0,948−1,438) p = 0.145 | 1,032 (0,832−1,281) p = 0.775 |
| | March | 8.6% (n = 207) | 8.3% (n = 3921) | 1,204 (0,983−1,474) p = 0.072 | 1,115 (0,904−1,375) p = 0.310 |
| | April | 7.6% (n = 183) | 7.7% (n = 3640) | 1,147 (0,931−1,413) p = 0.189 | 1,117 (0,899−1,386) p = 0.318 |
| | May | 6.9% (n = 166) | 8.0% (n = 3782) | 1,001 (0,809−1,239) p = 0.992 | 0,988 (0,792−1,232) p = 0.916 |
| | June | 7.9% (n = 191) | 8.5% (n = 4018) | 1,0849 (0,882−1,332) p = 0.442 | 1,052 (0,850−1,303) p = 0.641 |
| | July | 7.7% (n = 187) | 9.0% (n = 4265) | Reference | Reference |
| | August | 9.3% (n = 224) | 8.8% (n = 4192) | 1,219 (0,999−1,487) p = 0.051 | 1,202 (0,978−1,477) p = 0.08 |
| | September | 8.7% (n = 211) | 8.5% (n = 4032) | 1,194 (0,976−1,460) p = 0.085 | 1,178 (0,956−1,452) p = 0.125 |
| | October | 8.9% (n = 214) | 8.7% (n = 4111) | 1,187 (0,971−1,451) p = 0.094 | 1,123 (0,912−1,384) p = 0.274 |
| | November | 8.4% (n = 204) | 8.2% (n = 3900) | 1,193 (0,974−1,462) p = 0.089 | 1,128 (0,914−1,392) p = 0.263 |
| | December | 8.9% (n = 216) | 8.4% (n = 4008) | **1,229** (1,006-1,**502**) p = 0.044* | 1,134 (0,921−1,397) p = 0.235 |

*Reference – month of birth with the lowest disease prevalence determined in the initial crude estimate. Abbreviations: cOR- crude odds ratio; aOR- adjusted odds ratio; 95% CI – 95% confidence interval. Adjusted for month, sex, age, family allergy history, the presence of allergic disease of interest -AD, AR, asthma – in combinations. Statistically significant differences are highlighted in bold. *p < 0.05; **p < 0.0281 (with Benjamini Hochberg adjustment).*

The extent to which a specific month influenced the likelihood of developing AD was somewhat lower. For example, the odds of AD development among individuals born between August and February was significantly higher compared to those born in April (reference), with the strongest association observed in October (aOR, 1,169; 95% CI, 1,059−1,291), where it was 16% higher compared to April. The association observed for October remained significant following the adjustment for multiple comparisons (Table 3, Fig 3).

In the case of asthma, the association between the month of birth and the odds of asthma development was observed in January (cOR, 1.307; 95% CI, 1.072–1.594) and December (cOR, 1.229; 95% CI, 1.006–1.502) compared with July (reference). However, when the effects of all other variables were accounted for, all differences between months ceased to be statistically significant.

## Discussion

This is the first study in Russia to demonstrate that children born in October in Moscow face an elevated odd of atopic dermatitis, while children born in December, January, and February are more susceptible to allergic rhinitis. Our data showed that the association detected was independent of sex, age, family allergic history and combination of allergic diseases.

The reported estimates of AD, AR and asthma prevalence in the pediatric population vary significantly, due to demographic characteristics of different samples, regional variations, different methodologies of data collection etc. In Europe

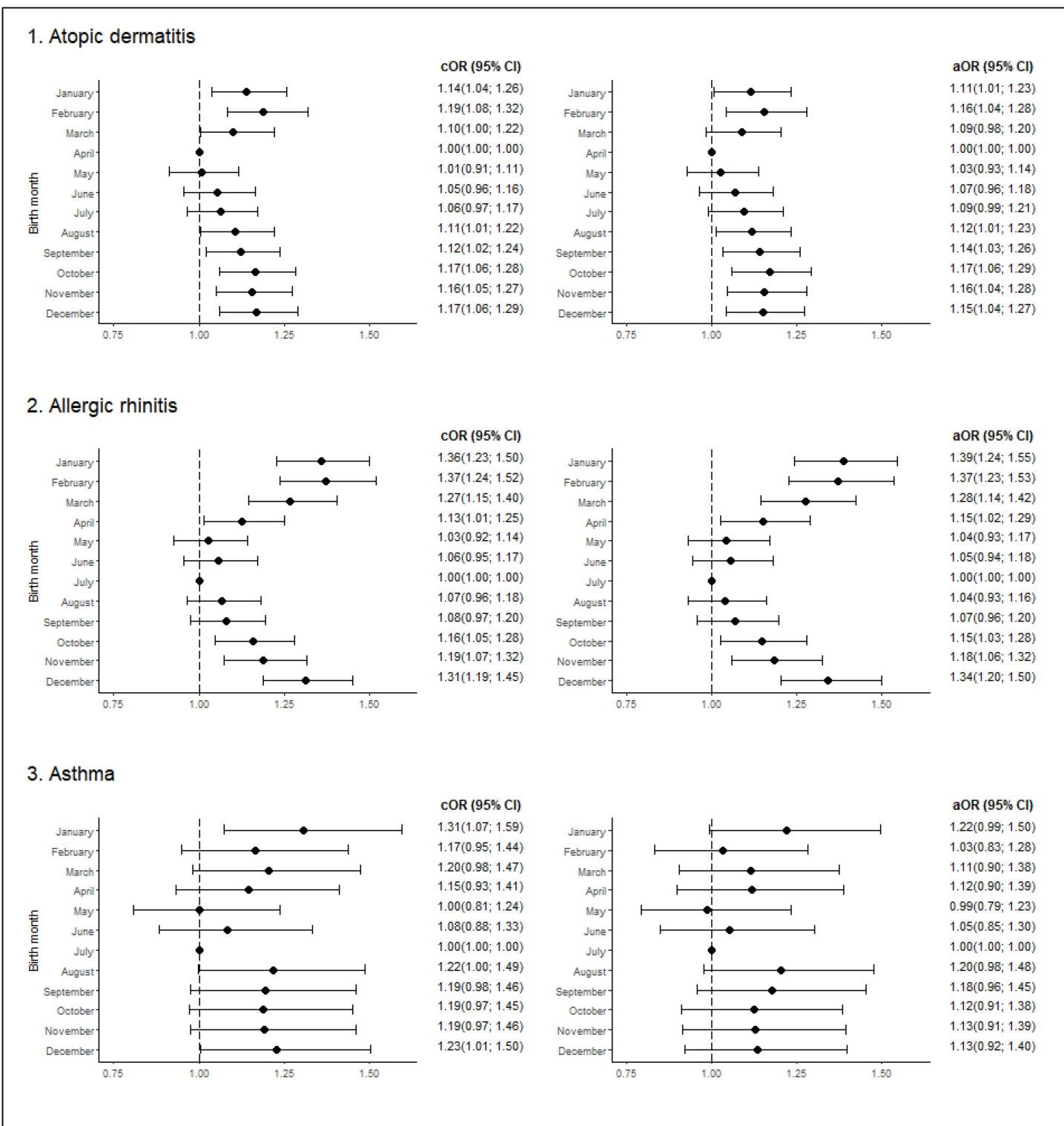

**Fig 3. Odds ratio and 95% CI for outcomes (development of allergic disease) by birth month.** *Reference – month of birth with the lowest disease prevalence determined in the initial crude estimate. Abbreviations: cOR- crude odds ratio; aOR- adjusted odds ratio; 95% CI – 95% confidence interval. Adjusted for month, sex, age, family allergy history, the presence of allergic disease of interest – AD, AR, asthma – in combinations.*

and the USA, the prevalence of AD in children is up to 20% [22]. In about 80% of patients with AD, the onset of the disease occurs early in their lives [23]; importantly, 10–30% of children with AD also have asthma and AR [24,25]. According to the epidemiological International Study of Asthma and Allergies in Childhood, ISAAC, AR symptoms are experienced, on average, by 8.5% of children aged 6–7 years and 14.6% of schoolchildren aged 13–14 years, asthma symptoms – in 11.8% children aged 6–7 years and 13.8% schoolchildren aged 13–14 years. [26–28]. Our study revealed that, compared with global trends, children residing in Moscow, Russia, presented high prevalences of AD (29.4%) and AR (23.2%), with a somewhat lower asthma prevalence of 4.84%. The differences we obtained may be associated with different methodologies for collecting clinical and anamnestic data, and different climatic, geographical, and socio-economic characteristics of the regions.

The family allergy history has been demonstrated to significantly increase the risk of developing allergic disease, whereas the effect of sex is ambiguous and depends on the child's age. For example, meta-analyses have shown that in infants and preschool children AD, asthma and AR are more prevalent among boys than among girls, whereas during puberty, these diseases become more prevalent among female patients [29–31]. In our study, the median age of the patients was 7 years, with male patients and children with a family allergy history being at greater odds of developing all three allergic diseases, which is consistent with the general trend. The association between demographic factors and disease development was the strongest for patients with asthma and the weakest for those with AD.

The associations between the birth season as a factor significantly influencing the child during the neonatal adaptation period, and the risk of developing allergic diseases, have been investigated in several studies [10−19]. To date, this is the first study of this nature in Russia. In this study of children residing in Moscow we revealed an association between the period and month of birth and the odds of developing AR and AD. For asthma, we did not find a similar significant association, which may be due to the later manifestation of asthma compared with AD and AR, and the consequent lower and/or more distant influence of the seasonal birth factor. The odds of AD development were significantly greater in children born between August and February, whereas the highest odds of AR development were observed for children born between October and April. The strongest association between the specific month of birth and the odds of AR development was established among children born in February, January and December. In the case of AD, the strongest association between the specific month of birth and the odds of AD development was established for children born in October. Notably, the season/month of birth was an independent criterion influencing the odds of diseases development, as the degree of association did not depend on demographic parameters, including sex, age, family allergic history, and overlapping allergic disease of interest. It can be hypothesized that these results can be explained by the Moscow climate characteristics. Moscow is a part of the central region of the Russian Federation, where the climate is intermediate between mild European and strongly continental Asian climates, with distinct seasonal patterns: long rainy autumns, moderately humid and cold winters, warm summers. The coldest months of the year are December, January and February, which have the highest associations with the odds of developing the allergic diseases in question. Among the factors determining the autumn-winter seasonality and having potential significance in the context of the odds of allergic disease, one can additionally highlight the low degree of solar activity typical for the Moscow region, high exposure to household allergens of house dust, high infectious load, etc. In addition, a seasonal increase in exposure to pet allergens in residential areas of Moscow may have a certain impact. This sensitization to cat and dog allergens appears to be extremely widespread among Russians [32]. The possible roles of decreased vitamin D levels in serum, increased cord blood IgE levels in winter, increased immune activity in general in winter months, including gene expression of several interleukins (IL-4, IL-5), etc., are actively discussed [19,33]. Cesarean section is an important known and common perinatal factor that most studies suggest may increase the odds of allergic diseases [34,35]. It could be assumed that the higher frequency of cesarean section in the fall and winter months could to some extent determine the association of the season of birth and the odds of AD and AR that we observed. However, in Russia, cesarean sections are performed only when strict indications exist and there are no data on the frequency of cesarean sections in different periods of the year. In an Iranian study, Nasiri R. et al.

reported that seasonal (monthly) variations of the weather (humidity and temperature) have a significant impression on preeclampsia prevalence, which is one of the important indications for cesarean section [36]. Based on time of conception the lowest prevalence of preeclampsia was found in winter and early spring, which corresponds to the lowest frequency of cesarean section in the fall and winter period. Thus, the effect of cesarean section on the association of season of birth and risk of allergic disease is ambiguous but extremely interesting and requires further study.

Our data on AD are consistent with the results of the majority of similar studies carried out in different countries, even those with less pronounced seasonality and warm winters. Yokomichi H. et al. in a study involving 100,304 Japanese children aged up to 3 years and born between 2011 and 2014, reported that the highest incidence of AD occurs in children born between October and December, and the lowest occurs between April and June [10]. Considering the effects of UV (photoperiod length) and ambient humidity on the skin barrier, the authors specifically assessed these factors. The combination of a short photoperiod and low humidity was shown to be associated with the highest incidence of AD. The results of another study by Japanese colleagues, which evaluated the associations of the season of birth with AD at different ages of the child up to 1 year of age, suggest a high risk of AD at 6 months in children born in autumn. This risk was particularly notable in boys whose mothers had allergic diseases [11]. Kuo CL et al. reported that in Taiwan, children born between October and December have a high risk of AD, with the highest risk occurring in December [12]. A systematic review (9 articles with 726,378 children aged 0–12 years) in Northern Hemisphere countries far from the equator reported a high incidence of AD among children born in the autumn and winter months [13].

In turn, the associations of the season of birth with the odds of asthma and AR vary across studies, which may be different from AD, and this finding is not as clearly consistent with our results. Thus, according to Hänninen R et al. in Finland, birth in the autumn-spring period was significantly associated with AR, and birth in the autumn or winter was significantly associated with asthma [14]. A similar population-based study in this region revealed that those born between January and June were at increased odds of asthma [15].

In a study of schoolchildren in Taiwan, it was noted that children born in winter (August to October) had a greater prevalence of asthma compared to children born in the spring (February to April), and no similar associations were found for AR and AD [16]. In Japan, Saitoh Y et al. analyzed the associations of birth month with sensitization to aeroallergens and the occurrence of allergic diseases among 755 schoolchildren aged 12–13 years. Sensitization to house dust mites was less common in children born between January and March, and sensitization to Japanese cedar pollen was significantly more common in children born between December and January. A significantly greater prevalence of asthma was observed among children born between November and December, for AR – between August and October. This study did not find an association between AD and the season of birth [17]. According to Knudsen T. et al. in Denmark, asthma and sensitization to house dust mites were more common in persons born in autumn, whereas AR and sensitization to pollen were not associated with the season of birth [18]. Overall, our data and the data from various studies regarding the association of season of birth with the odds of developing AR and asthma demonstrate heterogeneity, but no study showed an increased odds of AR and asthma among those born in the summer months. This observation probably supports the positive effect of warm climatic conditions among those born in the summer months on reducing the odds of developing allergic diseases.

Therefore, the seasonality factor is not homogeneous and includes many uncontrolled and controlled aspects that require further investigation. Available data on the effect of the season of birth on the odds of allergy in different regions can vary considerably, demonstrating specific geographic patterns, even within the same country, in addition, the size of the patient sample, research methodology, and adjustments for various factors may be of significant importance for these differences. Understanding the mechanisms and degree of influence of the season of birth on the odds of allergic diseases can certainly be useful in designing and interpreting the results of preventive interventions, optimal personalized prevention during pregnancy planning, as well as before and after the birth of the child. Specifically, based on our data, a potential strategy for preventing allergic diseases in children in Moscow could be planning pregnancy with the birth of

children in the spring-summer period between March and July to reduce the risk of AD and in the spring-autumn period between May and September to reduce the risk of AR, especially in cases of family allergy history. Regardless of the season of birth, important preventive measures include: breastfeeding, correcting and maintaining normal vitamin D levels in blood, moisturizing and maintaining the skin's barrier function, reducing the infectious load, introducing complementary foods in a timely manner, creating a supportive social and psychological environment, etc. Furthermore, the influence of seasonality on the risk of developing allergy should be considered when planning further epidemiological studies, inter-preting the data obtained, and developing prevention strategies based on them. Further epidemiologic studies are needed to accumulate evidence of the identified association of season and month of birth with allergic disease odds, and addi-tional research is needed to find new potential factors that may determine and/or strengthen this association. Such factors include, for example, increasing exposure and sensitization to epidermal allergens (cats, dogs) in autumn and winter, viral infections in the first year of life, such as COVID-19, the presence of children older than 2 years of age in the family (i.e., preschool and school age), which contribute to the infant's early exposure to respiratory viruses, the socioeconomic status of the family, which to some extent may offset already known factors (medication correction of vitamin D levels, travel to countries with warmer climates, monitoring of the child's health, etc.), conception season, use of in vitro fertilization, pre-mature birth, negative perinatal factors (prematurity, hypoxia, etc.). In addition, it is relevant to study not only the associa-tion of season and month of birth with the risk of allergic disease, but also the possible influence of season of birth on the severity and prognosis of the disease.

## Limitations

The limitations of the present study include the fact that the following factors were not taken into account in the analysis: sensitization, age of disease manifestation and diagnosis, disease severity, additional seasonality factors such as ambient air humidity and degree of solar activity, vitamin D levels, specificity and degree of allergen exposure in different periods of the year, infectious load, month of conception and gestational age and others. These factors may have potential impli-cations for the observed association between season of birth and allergic disease odds, for example, perhaps if vitamin D levels were taken into account in the data analysis, the highest odds of developing AD may have been observed in children born not in October, but in the cold winter months with minimal solar activity (December and February). The sam-ple included only children whose parents had previously completed a survey regarding family allergy history. In addition, parental responses may be subjective and not always accurate (e.g. parental recall bias). The analysis did not take into account the socioeconomic status of the family, which may influence the objectivity of parents' assessment of child health and other factors. The sample included only children living in Moscow and did not include children living in other regions of Russia.

## Conclusions

This is the first study conducted in Russia to evaluate the association between the month of birth and the odds of devel-oping allergic disease in children. In our work, we found that children born between August and February have greater odds of AD development among children living in Moscow, and that children born between October and April have greater odds of AR development. The strongest association between specific birth month and the odds of AD development was recorded among those born in October, the strongest association between specific birth months and the odds of AR was recorded among those born in the winter months (December, January and February). Our data showed that the associa-tion detected was independent of sex, age, family allergic history and combination of allergic diseases. Based on our data, an additional potential strategy for preventing allergic diseases among children living in Moscow could be planning preg-nancy with the birth of children in the spring-summer period between March and July to reduce the risk of AD and in the spring-autumn period between May and September to reduce the risk of AR, especially in cases of family allergy history. Further studies are needed to gather evidence for the observed associations of season and month of birth with disease

risk while considering additional factors and to better understand the mechanisms underlying this association. Furthermore, the possible association between season of birth and the risk of developing allergies should be taken into account when planning further epidemiological studies, interpreting the obtained data and developing preventive strategies based on them.

## Author contributions

**Conceptualization:** Alexander Pampura.

**Data curation:** Nikita Chikunov.

**Formal analysis:** Daria Dolotova.

**Visualization:** Olga Serebryakova.

**Writing – original draft:** Natalia Esakova.

**Writing – review & editing:** Alexander Pampura, Natalia Esakova, Daria Dolotova.

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
