## [Decision Letter · Decision Letter 0]

8 Apr 2025

Dear Dr. Esakova,

Thank you for submitting your manuscript to PLOS ONE. After careful consideration, we feel that it has merit but does not fully meet PLOS ONE’s publication criteria as it currently stands. Therefore, we invite you to submit a revised version of the manuscript that addresses the points raised during the review process.

We look forward to receiving your revised manuscript.

Kind regards,

Rajeev Singh

Academic Editor

PLOS ONE

Journal Requirements:

For additional information about PLOS ONE ethical requirements for human subjects research, please refer to http://journals.plos.org/plosone/s/submission-guidelines#loc-human-subjects-research .

3. Thank you for stating the following financial disclosure: [The study was sponsored by Moscow Center for Innovative Technologies in Healthcare and was funded by grant from the Moscow government [research project No. 0408-1]. 

Reviewers' comments:

Reviewer's Responses to Questions

**Comments to the Author**

1. Is the manuscript technically sound, and do the data support the conclusions?

Reviewer #1: Yes

Reviewer #2: Partly

2. Has the statistical analysis been performed appropriately and rigorously?

Reviewer #1: Yes

Reviewer #2: Yes

3. Have the authors made all data underlying the findings in their manuscript fully available?

Reviewer #1: Yes

Reviewer #2: Yes

4. Is the manuscript presented in an intelligible fashion and written in standard English?

Reviewer #1: Yes

Reviewer #2: Yes

Reviewer #1: This manuscript presents a novel and region-specific epidemiological study on the association between birth month and allergic diseases (AD, AR, and asthma) in children. The dataset size and analytical methods are commendable. This is the first study of its kind in Russia and adds valuable data to the international literature on seasonal and perinatal risk factors for allergy.

Major Comments:

The conclusion implies causality (e.g., "seasonality is an independent factor"). This wording should be revised to reflect an association rather than causation due to the observational nature of the study.

While asthma was included, its adjusted odds ratios were not statistically significant. This point should be addressed more clearly in the Discussion section.

Limitations such as parental recall bias, self-selection in the data collection process, and potential unmeasured confounding (e.g., vitamin D levels, exposure to specific allergens) should be more explicitly discussed.

Figures and tables should include more descriptive legends for better standalone comprehension.

Minor Comments:

Minor language edits are suggested, such as:

“requiring merits further investigation” → “which merits further investigation”

"In the Russian" → "In Russia"

Reviewer #2: Thank you for the opportunity to review this study on month of birth and allergic outcomes based in Moscow, Russia.

It is nice to see some Russian data on this topic and excellent that the authors have put it in the context of other findings particularly in the northern hemisphere with similar seasonality.

Some comments:

1. METHODS: The main issue with this study is that it potentially is affected by selection bias. The study population has been selected as children who went to a pediatric visit. Does this mean that the study population is only children who have some kind of illness or do all children in Russia go to pediatricians? I think more context needs to be supplied. In Western countries usually only children with very particular issues go to pediatricians, and often only if their parents can afford it. Therefore, there could be two layers of selection bias which effect the generalisability of these findings - only children with illnesses, and only wealthy people. Please put more context and description around these points. The fact that there are 50 000 children who go to pediatricians in only a 9 month period seems really large, so I think explanations about practices in Russia would help. Is there a way to include the general population in this study? if not, then it needs to be specified in the abstract, main findings, discussion that these findings are within a population of children with health issues and the issues with generalisability must be discussed in the Limitations section.

2. METHODS: Is there a way the authors can adjust for socioeconomic status which is an important confounder and potential source of selection bias in this study? if not, please discuss in limitations and its effect.

3. METHODS: Is it possible to adjust for month of conception or adverse perinatal outcomes such as gestational age? these could both be affecting the outcomes

4. METHODS: Related to point 1, please provide more information about what the EMIAS is and how data is collected.

5. METHODS: Was it the pediatricians who assigned the ICD-10 code from the visit?

6. METHODS: I think it would help if you provided inclusion and exclusion criteria to the study population. I am unsure if they only people included were those with allergy or whether the ICD 10 codes were used to define the outcomes of interest.

7. RESULTS: I am surprised that the children started at age 4 years, since AD, AR and asthma often show up in younger children. Please again provide context as to why this is.

8. RESULTS: The term 'odds' should be used instead of 'risk' eg in the abstract and in the results. Risk implies relative risk which is different to an odds ratio. Similarly use 'association' rather than 'correlation'.

9. RESULTS: Please refer to the formatting instructions of the journal for decimal places and inclusion of p values. Usual practice is to have 2 rather than 3 DP and to not include p values.

10. DISCUSSION: The first paragraph of the discussion should be a summary of the findings, please add this.

11. DISCUSSION: there needs to be more discussion about why there is heterogeneity between findings in this area, it is not enough to just say there is heterogeneity, some thoughts on why this is, and what your study adds are needed

12. DISCUSSION: an important undiscussed factor to explain the associations is perinatal factors associated with season which are also highly related to allergic outcomes. For instance, I am wondering if preterm births, low birth weight and caesarean sections are more likely to happen in winter which could explain the findings?

12. DISCUSSION: related, there needs to be more discussion about why the month of BIRTH is driving long term outcomes. that it could be affecting outcomes 6 months later as in some other studies makes sense, but month of birth on disease 4-18 years later seems a big claim. How do we know it is not the exposure in the first few months after birth or the month of conception or the exposures during pregnancy? I am trying to suggest to consider in the discussion what month of birth represents in a broader context. As the authors say, there are so many factors that could be influencing (confounding or mediating) these findings. Could the authors suggest given the breadth of literature on this topic which areas the research should branch into rather than more of the same?

**Do you want your identity to be public for this peer review?** For information about this choice, including consent withdrawal, please see our Privacy Policy

Reviewer #1: No

Reviewer #2: No

---

## [Author Response · Author response to Decision Letter 1]

4 Jun 2025

Reviewer #1

1. The conclusion implies causality (e.g., "seasonality is an independent factor"). This wording should be revised to reflect an association rather than causation due to the observational nature of the study.

The authors' response:

Dear reviewer! We have changed the wording (e.g., "seasonality is an independent factor") to reflect an association rather than causation.

2. While asthma was included, its adjusted odds ratios were not statistically significant. This point should be addressed more clearly in the Discussion section.

The authors' response:

Dear reviewer! The absence of a statistically significant association between asthma and month of birth is noted in the Discussion section.

3. Limitations such as parental recall bias, self-selection in the data collection process, and potential unmeasured confounding (e.g., vitamin D levels, exposure to specific allergens) should be more

explicitly discussed.

The authors' response:

Dear reviewer! We have included additional points in the limitation section.

4. Figures and tables should include more descriptive legends for better standalone comprehension.

The authors' response:

Dear reviewer! We have corrected the title of table 2.

5. Minor language edits are suggested, such as:

“requiring merits further investigation” → “which merits further investigation”

"In the Russian" → "In Russia"

The authors' response:

Dear reviewer! We have corrected minor language edits.

Reviewer #2

1. METHODS: The main issue with this study is that it potentially is affected by selection bias. The study population has been selected as children who went to a pediatric visit. Does this mean that the study population is only children who have some kind of illness or do all children in Russia go to pediatricians? I think more context needs to be supplied. In Western countries usually only children with very particular issues go to pediatricians, and often only if their parents can afford it. Therefore, there could be two layers of selection bias which effect the generalisability of these findings - only children with illnesses, and only wealthy people. Please put more context and description around these points. The fact that there are 50 000 children who go to pediatricians in only a 9 month period seems really large, so I think explanations about practices in Russia would help. Is there a way to include the general population in this study? if not, then it needs to be specified in the abstract, main findings, discussion that these findings are within a population of children with health issues and the issues with generalisability must be discussed in the Limitations section.

The authors' response:

Dear reviewer! In Russia, the basic medical care for children and adults is provided through compulsory health insurance, i.e. free of charge. Children may visit a pediatrician due to illness (e.g., often in the case of an acute viral infection or in connection with some specific complaints or diseases), or patients may come without complaints or diseases, e.g., to obtain school or sectional certificates, to undergo a checkup or vaccination. Therefore, the study population included both children with diseases and healthy children. In our study there are no such two levels of selection bias: children with diseases and only wealthy people. We have added additional information to the Methods section.

2. METHODS: Is there a way the authors can adjust for socioeconomic status which is an important confounder and potential source of selection bias in this study? if not, please discuss in limitations and its effect.

The authors' response:

Dear reviewer! Visiting a pediatrician in Russia is free of charge, so socioeconomic status is not an important factor and potential source of selection bias in this study. The EMIAS system does not take into account the socioeconomic status of the family. We have included this aspect in the limitation section.

3. METHODS: Is it possible to adjust for month of conception or adverse perinatal outcomes such as gestational age? these could both be affecting the outcomes.

The authors' response:

Dear reviewer! Unfortunately, we cannot make an adjustment for the month of conception and gestational age, since this information is not in the EMIAS system. We have included this aspect in the limitation section.

4. METHODS: Related to point 1, please provide more information about what the EMIAS is and how data is collected.

The authors' response:

Dear reviewer! We have added information about EMIAS to the Methods section.

5. METHODS: Was it the pediatricians who assigned the ICD-10 code from the visit?

The authors' response:

Dear reviewer! If a child visits a pediatrician with a disease, the pediatrician makes a diagnosis with diagnosis code according to ICD-10.

6. METHODS: I think it would help if you provided inclusion and exclusion criteria to the study population. I am unsure if they only people included were those with allergy or whether the ICD 10 codes were used to define the outcomes of interest.

The authors' response:

Dear reviewer! The main inclusion criteria for the formation of the database from EMIAS were: age under 18 years and a parentally completed questionnaire to identify family allergy history. We have specified these criteria in the methods section.

7. RESULTS: I am surprised that the children started at age 4 years, since AD, AR and asthma often show up in younger children. Please again provide context as to why this is.

The authors' response:

Dear reviewer! The median age of the children whose parents completed the questionnaire is given in the article. This age is not related to the onset of allergic disease, i.e. the data download only recorded the fact that the child had the disease in the medical record, which could have been at any year of life.

8. RESULTS: The term 'odds' should be used instead of 'risk' eg in the abstract and in the results. Risk implies relative risk which is different to an odds ratio. Similarly use 'association' rather than 'correlation'.

The authors' response:

Dear reviewer! We have changed the term 'risk' to 'odds', and the term 'correlation' to 'association'.

9. RESULTS: Please refer to the formatting instructions of the journal for decimal places and inclusion of p values. Usual practice is to have 2 rather than 3 DP and to not include p values.

The authors' response:

Dear reviewer! We checked the Submission Guidelines (https://journals.plos.org/plosone/s/submission-guidelines ) and found the following phrase in the Statistical Reporting section:

“P-values. Report exact p-values for all values greater than or equal to 0.001. P-values less than 0.001 may be expressed as p < 0.001, or as exponentials in studies of genetic associations.”

The practice of reporting p-values with three decimal places while presenting results in medicine has been highlighted in many statistical guidebooks and articles (https://pmc.ncbi.nlm.nih.gov/articles/PMC7642026/ , https://pmc.ncbi.nlm.nih.gov/articles/PMC8024217/ , https://pmc.ncbi.nlm.nih.gov/articles/PMC8005799/)

Additionally, the third decimal place can be very important, especially in case of adjustment for multiplicity, as can be seen from our tables (for example, p-values of 0.028 or 0.027 from Table 3, that are very close to the adjusted threshold). We would like to present our results to readers in the most transparent way.

10. DISCUSSION: The first paragraph of the discussion should be a summary of the findings, please add this.

The authors' response:

Dear reviewer! We have added the first paragraph to the discussion section.

11. DISCUSSION: there needs to be more discussion about why there is heterogeneity between findings in this area, it is not enough to just say there is heterogeneity, some thoughts on why this is, and what your study adds are needed

The authors' response:

Dear reviewer! Overall, our data and the data from various studies regarding the association of season of birth with the odds of developing AR and asthma demonstrate heterogeneity, but no study showed an increased odds of AR and asthma among those born in the summer months. This observation probably supports the positive effect of warm climatic conditions among those born in the summer months on reducing the odds of developing allergic diseases.

Available data on the effect of the season of birth on the odds of allergy in different regions can vary considerably, demonstrating specific geographic patterns, even within the same country, in addition, the size of the patient sample, research methodology, and adjustments for various factors may be of significant importance for these differences.

We have added this information to the discussion section.

12. DISCUSSION: an important undiscussed factor to explain the associations is perinatal factors associated with season which are also highly related to allergic outcomes. For instance, I am wondering if preterm births, low birth weight and caesarean sections are more likely to happen in winter which could explain the findings?

The authors' response:

Dear reviewer! Cesarean section is an important known and common perinatal factor that most studies suggest may increase the odds of allergic diseases. It could be assumed that the higher frequency of cesarean section in the fall and winter months could to some extent determine the association of the season of birth and the odds of AD and AR that we observed. However, in Russia, cesarean sections are performed only when strict indications exist and there are no data on the frequency of cesarean sections in different periods of the year. In an Iranian study, Nasiri R. et al. reported that seasonal (monthly) variations of the weather (humidity and temperature) have a significant impression on preeclampsia prevalence, which is one of the important indications for cesarean section. Based on time of conception the lowest prevalence of preeclampsia was found in winter and early spring, which corresponds to the lowest frequency of cesarean section in the fall and winter period. Thus, the effect of cesarean section on the association of season of birth and risk of allergic disease is ambiguous but extremely interesting and requires further study.

We have added this information to the discussion section.

13. DISCUSSION: related, there needs to be more discussion about why the month of BIRTH is driving long term outcomes. that it could be affecting outcomes 6 months later as in some other studies makes sense, but month of birth on disease 4-18 years later seems a big claim. How do we know it is not the exposure in the first few months after birth or the month of conception or the exposures during pregnancy? I am trying to suggest to consider in the discussion what month of birth represents in a broader context. As the authors say, there are so many factors that could be influencing (confounding or mediating) these findings. Could the authors suggest given the breadth of literature on this topic which areas the research should branch into rather than more of the same?

The authors' response:

Dear reviewer! Further epidemiologic studies are needed to accumulate evidence of the identified association of season and month of birth with allergic disease odds, and additional research is needed to find new potential factors that may determine and/or strengthen this association. Such factors include, for example, increasing exposure and sensitization to epidermal allergens (cats, dogs) in autumn and winter, viral infections in the first year of life, such as COVID-19, the presence of children older than 2 years of age in the family (i.e., preschool and school age), which contribute to the infant's early exposure to respiratory viruses, the socioeconomic status of the family, which to some extent may offset already known factors (medication correction of vitamin D levels, travel to countries with warmer climates, monitoring of the child's health, etc.), conception season, use of in vitro fertilization, premature birth, negative perinatal factors (prematurity, hypoxia, etc.). In addition, it is relevant to study not only the association of season and month of birth with the risk of allergic disease, but also the possible influence of season of birth on the severity and prognosis of the disease.

We have added this information to the discussion section.

---

## [Decision Letter · Decision Letter 1]

1 Sep 2025

Dear Dr. Esakova,

We look forward to receiving your revised manuscript.

Kind regards,

Vasuki Rajaguru, PhD

Academic Editor

PLOS ONE

Journal Requirements:

Reviewers' comments:

Reviewer's Responses to Questions

**Comments to the Author**

Reviewer #1: All comments have been addressed

Reviewer #3: (No Response)

2. Is the manuscript technically sound, and do the data support the conclusions?

Reviewer #1: (No Response)

Reviewer #3: Yes

3. Has the statistical analysis been performed appropriately and rigorously?

Reviewer #1: (No Response)

Reviewer #3: Yes

4. Have the authors made all data underlying the findings in their manuscript fully available?

Reviewer #1: (No Response)

Reviewer #3: Yes

5. Is the manuscript presented in an intelligible fashion and written in standard English?

Reviewer #1: (No Response)

Reviewer #3: Yes

Reviewer #1: (No Response)

Reviewer #3: Dear authors,

This study makes an important contribution to understanding seasonal birth effects on allergic diseases in Russian children. The large sample size, appropriate statistical methods, and novel geographic context are significant strengths. The findings generally align with global patterns showing increased allergy risk in autumn/winter births, likely mediated by vitamin D deficiency and other seasonal factors. Many methodological and statistical issues have been addressed in previous reviews. I have attached a file with some recommendations for improvement:

1. Abstract: Minor spelling corrections

I. "Crude odd ratio" should be "crude odds ratio" (line 31)

II. Consider stating the study period (2024) in methods

III. The phrase "elevated odd" (line 41) should be "elevated odds"

2. Introduction:

I. Lines 58-65: The sentence beginning with "Notably, the season factor itself is heterogeneous..." is overly complex and should be broken into shorter, clearer statements for better readability.

II. The connection between seasonal factors and specific mechanisms (vitamin D pathways, allergen exposure) could be included with more current evidence.

3. Methods: The manuscript does not address how missing data were managed. Authors should either: Describe the specific approach used for handling missing values (e.g., complete case analysis, multiple imputation) OR explicitly state if there were no missing values in the dataset

4. Results: The relatively low asthma prevalence (4.8%) compared to global ISAAC estimates (11.8-13.8%) requires more discussion.

5. Discussion: Recommended additional discussion of potential prevention strategies based on findings

6. Limitations: Missing discussion about generalizability beyond Moscow

7. Conclusion: Recommended to include clinical implications discussion for prevention strategies

Thank you

**Do you want your identity to be public for this peer review?** For information about this choice, including consent withdrawal, please see our Privacy Policy

Reviewer #1: No

Reviewer #3: **Yes: ** Durga Datta Chapagain

---

## [Author Response · Author response to Decision Letter 2]

16 Oct 2025

Response to Reviewer:

1. Abstract: Minor spelling corrections

I. "Crude odd ratio" should be "crude odds ratio" (line 31)

The authors' response:

Dear reviewer! We have corrected minor spelling edits. (line 31)

II. Consider stating the study period (2024) in methods

The authors' response:

Dear reviewer! The study period (2024) is included in methods. (line 27, line 79)

III. The phrase "elevated odd" (line 41) should be "elevated odds"

The authors' response:

Dear reviewer! We have corrected minor spelling edits. (line 41)

2. Introduction:

I. Lines 58-65: The sentence beginning with "Notably, the season factor itself is heterogeneous..." is overly complex and should be broken into shorter, clearer statements for better readability.

The authors' response:

Dear reviewer! The sentence was broken into shorter. (lines 59-62)

II. The connection between seasonal factors and specific mechanisms (vitamin D pathways, allergen exposure) could be included with more current evidence.

The authors' response:

Dear reviewer! We have added additional information to the Introduction section. (lines 62-67)

3. Methods: The manuscript does not address how missing data were managed. Authors should either: Describe the specific approach used for handling missing values (e.g., complete case analysis, multiple imputation) OR explicitly state if there were no missing values in the dataset

The authors' response:

Dear reviewer! We excluded all cases in which at least one of the study variables was missing, so there were no missing values in the dataset. We have added additional information to the Methods section. (lines 99-100)

4. Results: The relatively low asthma prevalence (4.8%) compared to global ISAAC estimates (11.8-13.8%) requires more discussion.

The authors' response:

Dear reviewer! Our study revealed that, compared with global trends, children residing in Moscow, Russia, presented high prevalences of AD (29.4%) and AR (23.2%), with a somewhat lower asthma prevalence of 4.84%. The differences we obtained may be associated with different methodologies for collecting clinical and anamnestic data, and different climatic, geographical, and socio-economic characteristics of the regions. We have added additional information to the Discussion section. (lines 241-243)

5. Discussion: Recommended additional discussion of potential prevention strategies based on findings

The authors' response:

Dear reviewer! Specifically, based on our data, a potential strategy for preventing allergic diseases in children in Moscow could be planning pregnancy with the birth of children in the spring-summer period between March and July to reduce the risk of AD and in the spring-autumn period between May and September to reduce the risk of AR, especially in cases of family allergy history. Regardless of the season of birth, important preventive measures include: breastfeeding, correcting and maintaining normal vitamin D levels in blood, moisturizing and maintaining the skin's barrier function, reducing the infectious load, introducing complementary foods in a timely manner, creating a supportive social and psychological environment, etc. Furthermore, the influence of seasonality on the risk of developing allergy should be considered when planning further epidemiological studies, interpreting the data obtained, and developing prevention strategies based on them. We have added additional information to the Discussion section. (lines 340-350)

6. Limitations: Missing discussion about generalizability beyond Moscow

The authors' response:

Dear reviewer! The sample included only children living in Moscow and did not include children living in other regions of Russia. We have added information about this limitation. (lines 376-378)

7. Conclusion: Recommended to include clinical implications discussion for prevention strategies

The authors' response:

Dear reviewer! Based on our data, an additional potential strategy for preventing allergic diseases among children living in Moscow could be planning pregnancy with the birth of children in the spring-summer period between March and July to reduce the risk of AD and in the spring-autumn period between May and September to reduce the risk of AR, especially in cases of family allergy history. Further studies are needed to gather evidence for the observed associations of season and month of birth with disease risk while considering additional factors and to better understand the mechanisms underlying this association. Furthermore, the possible association between season of birth and the risk of developing allergies should be taken into account when planning further epidemiological studies, interpreting the obtained data and developing preventive strategies based on them. We have added additional information to the Conclusion section. (lines 388-397)

---

## [Editor Report · Decision Letter 2]

20 Oct 2025

Birth month as a risk factor of allergic diseases: analysis of database of about 50 thousand children

PONE-D-25-05996R2

Dear Dr. Natalia Esakova,

We’re pleased to inform you that your manuscript has been judged scientifically suitable for publication and will be formally accepted for publication once it meets all outstanding technical requirements.

Kind regards,

Vasuki Rajaguru, PhD

Academic Editor

PLOS ONE

Additional Editor Comments (optional):

All the required revisions are amended.

---

## [Editor Report · Acceptance letter]

PONE-D-25-05996R2

PLOS ONE

Dear Dr. Esakova,

I'm pleased to inform you that your manuscript has been deemed suitable for publication in PLOS ONE. Congratulations! Your manuscript is now being handed over to our production team.

Kind regards,

on behalf of

Dr. Vasuki Rajaguru

Academic Editor

PLOS ONE